# Access to HIV-prevention in female sex workers in Ukraine between 2009 and 2017: Coverage, barriers and facilitators

**Natasha Blumer**[1,2¤]*, **Lisa M. Pfadenhauer**[1,2], **Jacob Burns**[1,2]

**1** Institute for Medical Information Processing, Biometry and Epidemiology – IBE, Ludwig-Maximilians-Universität München, Munich, Germany, **2** Pettenkofer School of Public Health Munich, Munich, Germany

¤ Current address: Institute for Medical Information Processing, Biometry and Epidemiology - IBE, Chair of Public Health and Health Services Research Pettenkofer School of Public Health (PSPH), LMU Munich, Munich, Germany

* natasha.blumer@posteo.de

## Abstract

The provision of comprehensive prevention services is vital for reducing the high burden of HIV amongst Ukrainian female sex workers (FSWs). To identify barriers and facilitators that influence access to HIV prevention amongst this population between 2009 and 2017, we developed a literature-informed conceptual framework and conducted a document analysis to identify the components of the Ukrainian prevention package (PP). Using the Integrated Bio Behavioural Surveillance Surveys, we then conducted descriptive analyses to explore PP coverage from 2009 to 2017 and the influence of factors, identified by our conceptual framework. After increasing over four years, a drop in PP coverage was observed from 2013 onwards. Being a client of a non-governmental organisation, street and highway solicitation, non-condom use, and knowledge of HIV may influence access to HIV prevention in the Ukrainian context. Future interventions should consider barriers and facilitators to HIV prevention and the multiple structural levels on which they operate.

## Introduction

Although the global HIV incidence rate has decreased, Ukraine has developed one of the most severe HIV epidemics in Europe and Central Asia [1]. As in most countries, the HIV epidemic in Ukraine is concentrated amongst populations that are at heightened risk of contraction, designated 'key populations'. These include people who inject drugs, men who have sex with men, transgender people, incarcerated persons and sex workers (SWs) [2–4].

Globally, female sex workers (FSWs) bear a disproportionately high burden of the HIV pandemic, with a global prevalence of over 10% [5]. FSWs are 13.5 times more likely to contract HIV than of women of the same reproductive age [6]. In Ukraine, the annual incidence rate lies at 0.56 per 1000 uninfected SWs and the prevalence at 5.2% within an estimated population of 80.000 SWs [3]. An array of determinants increase the vulnerability of a FSW to HIV, including gender-based violence (GBV) [7, 8], poverty [9, 10], multiple sexual partners [11], criminalisation [12–15], stigma [16, 17] and substance abuse [18]. Recent literature has posited

**Data Availability Statement:** The paper analyses the data from studies implemented by the International Charitable Foundation "Alliance for Public Health" (Alliance). The holders of the data

are Alliance and the Public Health Centre of the Ukrainian Ministry of Health (PHC), and both data holders practice restricted data sharing. Researchers wishing to access original datasets are recommended to submit an official letter to the Director General of the PHC via info@phc.org.ua and to both the Senior Program Officer for Research and Evaluation, Oksana Kovtun (kovtun@aph.org.ua) and the Program Director of "M&E-related Technical Assistance and Improved Data application in HIV (METIDA)", Tetyana Salyuk (salyuk@aph.org.ua) of Alliance with a description of their intent. The authors did not have any special access privileges that others would not have.

**Funding:** The author(s) received no specific funding for this work.

**Competing interests:** Data collection for this study was undertaken while NB was affiliated to PHC. I have read the journal's policy, and the authors of this manuscript have the following competing interests and confirm that this does not alter our adherence to PLOS ONE policies on sharing data and materials. All opinions presented in this manuscript belong to the authors alone, and not any institution to which they are or were affiliated, namely, the Alliance, the PHC or the funder of the data collection, the Global Fund to Fight AIDS, Tuberculosis and Malaria. The authors declare that they have no competing interests.

that determinants of HIV operate on different structural levels, ranging from the macrostructural to the individual level [13, 14].

Although a multitude of prevention interventions have been designed to reduce the impact of HIV amongst key populations, it has been established that no single prevention intervention is sufficient to contain the spread of the disease and that a multi-pronged, multi-level approach is necessary [19–21]. In 2014, the World Health Organisation (WHO) published guidelines consolidating all existing recommendations relevant to the prevention of HIV amongst key populations, including SWs. These guidelines outline the components of a comprehensive, evidence-based package of interventions, which has been found to be acceptable and effective for the global SW community. The package has two parts: essential health care sector interventions and essential strategies for an enabling environment. Health care interventions include components such as comprehensive condom and lubricant provision and behavioural interventions (e.g. consultation with social workers), whilst strategies addressing community empowerment and addressing violence fall beneath the enabling environment section [4]. As one of the five prevention pillars, the United Nations General Assembly prevention targets emphasised the importance of this comprehensive prevention package (PP), by urging countries to ensure that 90% of people at risk of HIV infection have access to the PP by 2020 [22]. The Alliance of Public Health (Alliance) has been distributing HIV prevention amongst key populations in Ukraine since 2003, which was first coined the PP in 2007, and has since been scaled up countrywide [23].

Despite these efforts, only 46% and 48% of Ukrainian SWs accessed the PP in 2015 and 2017 respectively [24, 25]. Recent literature has identified that FSWs encounter barriers on multiple structural levels, which impede their access to HIV prevention services [21, 26]. In order to improve the intervention and its implementation in Ukraine, further investigation is necessary to determine the factors that hinder FSWs from accessing the PP.

The aim of this study was to therefore explore which factors influenced the access that Ukrainian FSWs had to the PP between 2009 and 2017. More specifically, our four research objectives were 1) to develop a conceptual framework illustrating the factors that influence FSWs access to prevention and the structural levels they operate on 2) to identify which components were included in the Ukrainian PP for SWs, 3) to measure the coverage of the PP in Ukraine amongst the FSW population and finally, 4) to describe and explore the factors associated with the access of Ukrainian FSW to the PP over this nine-year period.

## Method

We undertook multiple steps to address our four research objectives, as outlined in Fig 1. We began by developing a conceptual framework, informed by a literature review (objective 1). We then conducted a document analysis, which served to describe the Ukrainian PP and its components (objective 2). The questionnaires from five Ukrainian Integrated Bio Behavioural Surveillance Surveys (IBBS) were then reviewed to identify items that corresponded to both the factors identified in the conceptual framework and the components identified in the document analysis. These then served as our independent variables and outcome variables, respectively and were used to conduct exploratory analyses. This included calculating descriptive statistics to measure the coverage of the PP amongst the Ukrainian FSW population (objective 3) and a descriptive longitudinal comparison between FSWs that did and did not access prevention, to explore which factors influenced access to PP between 2009 and 2017 (objective 4). As no information about the gender of Ukrainian SWs is available in the IBBS data prior to 2015 and that in 2017, 97.72% of Ukrainian SWs were female (2.14% male and 0.14% transgender), we have referred to SWs as FSWs when referring to IBBS data, unless stated otherwise.

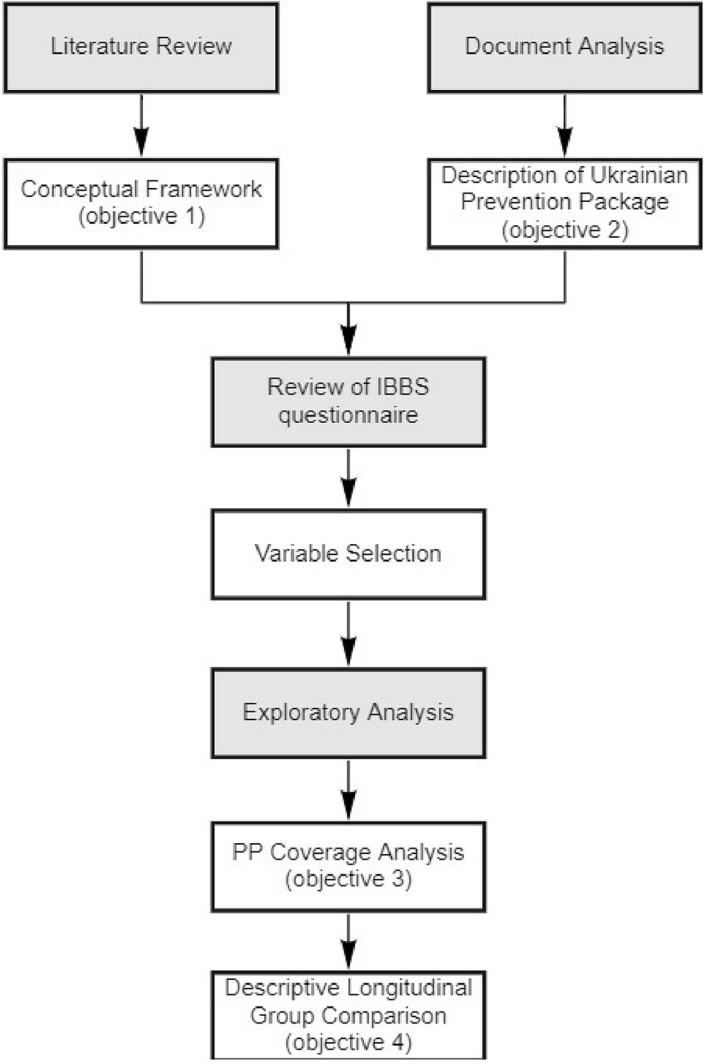

**Fig 1. Flowchart of methodological approach.**

## Conceptual framework

In an iterative process, NB thematically grouped specific factors associated with access to HIV prevention, which were identified in a literature review. The literature review method, corresponding protocol, and results are available in the S1 Appendix. Uncertainties about attributing factors to a certain thematic cluster were discussed with JB and LMP on an ongoing basis. To structure these thematic clusters, we adopted the framework structure developed by Shannon and colleagues, which visualises the determinants of HIV for the global population of FSWs. This framework identifies six structural levels that HIV determinants operate on, including the macrostructural, sex worker community organisation, work environment, sexual networks and patterning, sex worker and client individual level [13, 14]. As our literature review identified that factors associated with the access to HIV prevention also operate on different structural levels, we considered the structure of this framework appropriate and populated it with our thematic clusters accordingly (S1 Table).

## Document analysis and description of the Ukrainian PP

Our second research objective aimed to identify components offered as part of the Ukrainian PP between 2008 and 2017. To do so, we conducted a document analysis of all annual general reports published by Alliance over this time period [24, 25, 27–34]. With substantial financial support by of the Global Fund to Fight AIDS, Tuberculosis and Malaria (TGF), Alliance has been responsible for the distribution of HIV prevention amongst SWs in Ukraine since 2003, and each report provides a detailed list of components included in the PP [35]. These components were extracted from the English version of each annual report. After extraction, we grouped the components of similar nature into categories to gain a clearer picture of the composition of the PP over the years. For a detailed list of the categories and associated components, see S2 Table.

## Review of IBBS questionnaires and variable selection

We reviewed the questionnaires of five Integrated Bio Behavioural Surveillance Surveys (IBBS) to identify items that corresponded to both our outcome and independent variables, which were drawn from the results of the conceptual framework and document review, respectively. The IBBS are cross-sectional studies that were administered amongst Ukrainian SWs in 15 cities in 2009, 25 cities in 2011, 27 cities in 2013 and 2015 and 32 cities in 2017. The IBBS were developed, piloted and administered by Alliance under the financial support of the TGF in 2009, 2011 and 2013 and then from the The U.S. Centers for Disease Control and Prevention (CDC) in 2015 and 2017. The Public Health Centre (PHC) of the Ukrainian Ministry of Health are now the holders of the data. Ethical approval was granted by the Institutional Review Board (IRB) at the Ukrainian Institute on Public Health Policy (Kiev, Ukraine) and CDC. The surveys were administered as face-to-face interviews lasting approximately thirty minutes. The interview was followed by a voluntary rapid HIV tests, details of rapid tests can be found in the Alliance analytic reports [25, 28, 30, 32, 34]. Participants were considered eligible if they were fourteen or older, had received money, goods or services in exchange for sex within the last six months and consented to participate in the study. Three sampling strategies were employed for recruiting participants: respondent-driven-sampling (RDS), time-location-sampling (TLS) and key-informant sampling was employed in 2013, no follow up was conducted. Recruitment continued until sample size was reached. Sample size was calculated for each study site using the specific HIV prevalence rate as determined by the previous IBBS result. Further details about the study design of each year can be found in the analytic reports of Alliance [25, 28, 30, 32, 34]. All five IBBS surveys were translated from Ukrainian into English by an associate professor of Slavic languages at the Donetsk National Technical University.

**Independent variables.** To identify our independent variables, namely, *factors associated with access to HIV prevention*, we drew upon our conceptual framework (objective 1) and then screened each IBBS questionnaire for a corresponding item. The single most fitting item was selected for each independent variable.

We support the emphasis that Shannon and colleagues place upon factors that operate at the macrostructural level when designing prevention policies and schemas. As all IBBS questionnaires measure individual behaviour, we were unable to extract items operating on the macrostructural level, and hence, resorted to designating the most fitting item at the individual level to these macrostructural factors. The exact item used for each independent variable can be found in S4 Table.

**Outcome variable.** *Receipt of the Ukrainian PP* was our outcome of interest; hence we drew upon the findings of our document analysis. As there is no universally accepted way to measure PP receipt, we created two definitions, including "basic PP receipt" and "extended PP

receipt". We initially used Alliance's definition for basic PP receipt which consists of "the receipt of at least one consultation from a social worker and the receipt of at least one male condom over a six month period" [36]. However, due to limitations in the IBBS data, we altered this to "the receipt of at least one male condom and any form of HIV testing over a 12-month period".

For extended PP receipt, we adopted the Joint United Nations Programme on HIV/AIDS' (UNAIDS) [37] definition. According to UNAIDS, PP receipt is defined as, "access to at least two HIV prevention interventions within the last three months". As the time frame referenced in many IBBS question was twelve months, we adapted this definition by increasing the three-month period to twelve months. Thus, we defined extended PP receipt as, "access to at least two HIV prevention interventions within the last twelve months". The number and specific type of prevention services included in each year's extended PP, alongside the corresponding IBBS item, can be found in S3 Table.

## Exploratory analyses: PP coverage and descriptive longitudinal group comparison

We conducted descriptive statistics to measure the coverage of the basic and extended PP receipt between 2009 and 2017 (objective 3). We did so by calculating the proportion of FSWs that received the PP per year according to the definition for basic PP receipt and extended PP receipt. These results enabled us to undertake a descriptive longitudinal group comparison to compare the FSWs who did and did not receive the PP, with regard to the various independent variables (objective 4). Specifically, we calculated proportions for binary variables and means for continuous variables, stratified by those who received the basic PP and the extended PP and those who did not (Table 1). All variables were screened for missing data before undergoing descriptive analyses. For all but one variable assessed, the proportion of missing data was low, ranging from 0% to 6.7%. For the item regarding HIV status, a large proportion of the data, 54.3% was missing. We excluded missing data from the analysis. All analyses were conducted using the statistical software R (3.6.0) [38].

## Results

### Conceptual framework (objective 1)

Many factors associated with access to HIV prevention were presented in multiple studies, identified in our literature review. As these factors were largely aligned, we clustered factors with similar themes to better structure our conceptual framework. For example, three specific factors found to influence access to HIV prevention–knowledge of where to access prevention [9], knowledge of HIV transmission routes [39–41] and the self-perception of being at high risk of HIV infection [42]–were grouped into one broader thematic cluster, knowledge of HIV. After discussing the conceptual framework with staff members from the PHC, one thematic cluster, sex worker- family was excluded from the conceptual framework, as the specific factors were applicable to a generalised epidemic (HIV prevalence of ≥1% amongst the general population) and were not applicable in the Ukrainian context, where the epidemic is classified as a concentrated or mixed epidemic [43]. Our conceptual framework illustrates that most thematic clusters which influence access to HIV prevention operate at the macrostructural level, as seen in Fig 2.

### Document analysis and description of the Ukrainian PP (objective 2)

A total of 32 categories describing prevention components emerged from the document analysis of the Alliance annual general reports from 2008 to 2017 (see Table 2). Many components

**Table 1. Barriers and facilitators to accessing HIV prevention stratified by receipt or non-receipt of basic PP between 2009 and 2017.**

| | | 2009 (%) | 2011 (%) | 2013 (%) | 2015 (%) | 2017 (%) |
|---|---|---|---|---|---|---|
| Macrostructural | | | | | | |
| Socio-cultural | | | | | | |
| *Level of HIV stigma amongst the population* | | | | | | |
| Belief that HIV testing is personally not available | Received | 0.44 | 0.32 | 0.20 | 0.34 | 24.05 |
| | Not Received | 9.28 | 8.37 | 5.19 | 3.78 | 25.49 |
| Fear of HIV status being exposed[1] | Received[1] | **53.85**[1] | **9.09**[1] | - | **0.05**[1] | **1.19**[1] |
| | Not Received[1] | **31.43**[1] | **17.63**[1] | **0.86**[1] | **0.58**[1] | **3.76**[1] |
| No desire to know HIV test result[1] | Received[1] | 8.70[1] | 10.00[1] | 0.69[1] | - | 0.37[1] |
| | Not Received[1] | 8.23[1] | 10.63[1] | 1.03[1] | 0.18[1] | 2.76[1] |
| Geographical | | | | | | |
| *Location of HIV prevention service centres* | | | | | | |
| Inconvenient location of prevention service centres[1] | Received[1] | - | - | 0.04[1] | - | 0.59[1] |
| | Not Received[1] | 1.79[1] | 4.46[1] | 0.51[1] | 0.13[1] | 1.33[1] |
| *Migration associated with FSW* | | | | | | |
| History of travel for FSW purposes | Received | 21.98 | 16.28 | 12.33 | 7.85 | 6.98 |
| | Not Received | 15.50 | 14.18 | 16.20 | 7.52 | 10.71 |
| Health-Related Policy | | | | | | |
| *Behaviour of health care staff* | | | | | | |
| Dissatisfaction with staff attitude[1] | Received[1] | **7.69**[1] | **18.18**[1] | - | - | **0.15**[1] |
| | Not Received[1] | **3.57**[1] | **7.59**[1] | **0.30**[1] | **0.18**[1] | **0.19**[1] |
| *Functional hours of health care facility* | | | | | | |
| Inappropriate opening hours[1] | Received[1] | **7.69**[1] | **18.18**[1] | **0.08**[1] | - | **1.19**[1] |
| | Not Received[1] | **2.14**[1] | **6.25**[1] | **0.47**[1] | **0.09**[1] | **0.73**[1] |
| Economic | | | | | | |
| *National financial scheme for HIV prevention* | | | | | | |
| Belief that HIV prevention is too costly[1] | Received[1] | **7.69**[1] | **18.18**[1] | - | **1.61**[1] | **0.97**[1] |
| | Not Received[1] | **18.57**[1] | **15.62**[1] | **1.11**[1] | **2.80**[1] | **2.57**[1] |
| *National support schemas for women in the workforce* | | | | | | |
| FSW is the sole form of employment | Received | **40.00** | **47.45** | **41.77** | **58.92** | **60.73** |
| | Not Received | **33.80** | **43.01** | **39.95** | **42.22** | **51.52** |
| Community Organisation | | | | | | |
| Sex Worker Collectivisation | | | | | | |
| *Collective agency of FSW community* | | | | | | |
| Client of an NGO | Received | **82.86** | **94.14** | **95.51** | **97.03** | **94.43** |
| | Not Received | **14.69** | **20.49** | **24.82** | **25.18** | **15.21** |
| Work Environment | | | | | | |
| Physical | | | | | | |
| *Soliciting setting* | | | | | | |
| Street | Received | **23.30** | **27.33** | **15.17** | **15.29** | **17.45** |
| | Not Received | **10.96** | **17.46** | **14.81** | **10.94** | **8.82** |
| Highway | Received | **35.05** | **22.69** | **28.43** | **26.27** | **19.75** |
| | Not Received | **20.98** | **17.64** | **19.07** | **16.18** | **8.33** |
| Hotel | Received | 4.95 | 4.29 | 3.81 | 2.92 | 4.53 |
| | Not Received | 6.80 | 3.49 | 3.14 | 2.58 | 6.30 |
| Affluent settings (e.g. cafes, bars, modelling agencies) | Received | **10.66** | **13.62** | **11.27** | **12.37** | **10.62** |
| | Not Received | **22.81** | **17.36** | **18.68** | **16.90** | **20.05** |
| Social | | | | | | |

*(Continued)*

**Table 1.** (Continued)

| | | 2009 (%) | 2011 (%) | 2013 (%) | 2015 (%) | 2017 (%) |
|---|---|---|---|---|---|---|
| *Gender-based violence at workplace* | | | | | | |
| History of gender violence whilst providing sexual services | Received | - | **51.24** | **51.44** | **52.34** | **43.06** |
| | Not Received | - | **41.84** | **47.54** | **37.68** | **38.39** |
| Experience of gender violence from client[2] | Received[2] | - | 74.91[2] | 45.09[2] | 43.71[2] | 34.00[2] |
| | Not Received[2] | - | 72.74[2] | 42.69[2] | 30.60[2] | 29.95[2] |
| Experience of gender violence from a police officer[2] | Received | - | 32.04[2] | 12.94[2] | 7.55[2] | 7.80[2] |
| | Not Received | - | 24.96[2] | 8.57[2] | 5.16[2] | 7.66[2] |
| **Interpersonal dynamic** | | | | | | |
| Sex worker- client | | | | | | |
| *Condom use* | | | | | | |
| Sex without a condom with last client | Received | **4.29** | **3.79** | **1.78** | **4.24** | **3.12** |
| | Not Received | **13.82** | **8.77** | **4.50** | **8.63** | **6.71** |
| Pressure from client to have sex without a condom | Received | 23.08 | 48.81 | 0.81 | 1.66 | 1.63 |
| | Not Received | 33.86 | 38.62 | 1.80 | 4.09 | 3.30 |
| *Number of clients* | | | | | | |
| Number of clients in last 24 hours[+] | Received[+] | 2.26[+] | 2.24[+] | 2.11[+] | 1.87[+] | 2.42[+] |
| | Not Received[+] | 1.62[+] | 1.61[+] | 1.88[+] | 1.61[+] | 2.20[+] |
| Sex worker–non-partner | | | | | | |
| *Partner's knowledge of occupational status* | | | | | | |
| Partner knowledge of FSW status | Received | **40.11** | **38.39** | **17.83** | **18.52** | **15.81** |
| | Not Received | **35.44** | **31.43** | **12.56** | **11.61** | **8.33** |
| *Gender-based violence* | | | | | | |
| Experience of gender violence from intimate partner[2] | Received[2] | - | **10.39[2]** | **4.08[2]** | **2.49[2]** | **2.90[2]** |
| | Not Received[2] | - | **13.71[2]** | **3.51[2]** | **1.69[2]** | **3.17[2]** |
| *Partnership status* | | | | | | |
| Living with a partner | Received | 35.38 | 28.28 | 28.55 | 33.09 | 30.51 |
| | Not Received | 31.21 | 23.63 | 23.83 | 29.05 | 23.57 |
| **Sex Worker Individual** | | | | | | |
| Behavioural | | | | | | |
| *Knowledge of HIV* | | | | | | |
| Knowledge of HIV transmission routes (all questions correct) | Received | **64.07** | **67.88** | **60.09** | **59.55** | **57.76** |
| | Not Received | **55.63** | **51.25** | **55.68** | **57.38** | **50.46** |
| Knowledge of where to access HIV prevention | Received | **98.46** | **99.59** | **99.51** | **99.17** | **98.74** |
| | Not Received | **81.80** | **84.78** | **90.44** | **90.57** | **86.42** |
| *History of substance abuse* | | | | | | |
| Drug Injecting | | - | - | - | - | - |
| | | - | - | - | - | - |
| History of sharing needles | Received | 24.32 | 4.41 | 1.82 | 13.65 | 7.57 |
| | Not Received | 14.52 | 19.33 | 1.41 | 5.65 | 5.09 |
| *Work life balance* | | | | | | |
| Workdays per week[+] | Received[+] | 4.12[+] | 4.15[+] | 4.37[+] | 4.43[+] | 4.27[+] |
| | Not Received[+] | 2.97[+] | 3.22[+] | 4.02[+] | 3.77[+] | 3.87[+] |
| *Duration in sex work* | | | | | | |
| Age of FSW debut[+] | Received[+] | 20.28[+] | 21.24[+] | 21.65[+] | 21.77[+] | 23.16[+] |
| | Not Received[+] | 20.59[+] | 21.08[+] | 21.41[+] | 21.22[+] | 22.11[+] |
| Age of sexual debut[+] | Received[+] | 15.99[+] | 15.87[+] | 15.79[+] | - | 16.40[+] |
| | Not Received[+] | 15.92[+] | 15.99[+] | 15.89[+] | - | 16.17[+] |

*(Continued)*

**Table 1.** (Continued)

| | | 2009 (%) | 2011 (%) | 2013 (%) | 2015 (%) | 2017 (%) |
|---|---|---|---|---|---|---|
| *Level of education* | | | | | | |
| Higher education (tertiary education) | Received | 19.49 | 22.10 | 17.91 | 24.56 | 22.35 |
| | Not Received | 21.43 | 23.24 | 21.73 | 27.20 | 29.51 |
| Biological | | | | | | |
| *HIV factors* | | | | | | |
| HIV positive status | Received | 13.75 | 8.57 | 3.80 | 3.17 | 2.08 |
| | Not Received | 8.82 | 8.03 | 1.54 | 3.11 | 2.44 |
| *STI factors* | | | | | | |
| Diagnosis of hepatitis C within the last 12 months | Received | 5.93 | 6.18 | 8.98 | 7.16 | 6.38 |
| | Not Received | 4.46 | 4.13 | 4.59 | 4.54 | 3.81 |
| Diagnosis of hepatitis B within the last 12 months | Received | 2.97 | 1.94 | 2.39 | 2.24 | 2.30 |
| | Not Received | 1.54 | 1.07 | 1.41 | 1.51 | 2.41 |
| *Age* | | | | | | |
| Age[+] | Received[+] | 27.73[+] | 28.11[+] | 28.90[+] | 29.84[+] | 31.58[+] |
| | Not Received[+] | 27.15[+] | 27.28[+] | 28.09[+] | 27.98[+] | 28.92[+] |
| *Reproductive status* | | | | | | |
| Presence of dependents | Received | 51.10 | 60.89 | 43.39 | 44.01 | 46.70 |
| | Not Received | 47.44 | 50.36 | 51.95 | 52.98 | 57.90 |
| Number of dependents[+] | Received[+] | 1.94[+] | 1.83[+] | 1.91[+] | 1.03[+] | 0.98[+] |
| | Not Received[+] | 1.82[+] | 1.79[+] | 1.75[+] | 0.82[+] | 0.59[+] |
| Client individual | | | | | | |
| Biological | | | | | | |
| *Age* | | | | | | |
| Age ($\leq$ 35 years) | Received | 36.92 | 36.58 | 44.22 | 33.95 | 28.80 |
| | Not Received | 42.47 | 40.59 | 43.38 | 43.34 | 34.15 |

Bolded text are variables that are described in the results.

[1] Sub-question of "I believe that HIV testing is not available for me personally".

[2] Sub-question of "History of gender violence whilst providing sexual services'.

[+] Mean.

included in the PPs could be grouped into broader categories. For example, testing for chlamydia, assisted testing for syphilis and examination for sexually transmitted diseases (STI) were coded as the overarching category, 'any form of STI testing'. A list of specific components within each overarching prevention category can be found in S2 Table.

The number of components included in the Ukrainian PP increased from 2008 until 2013, when the highest number of components (n = 24) was included in both 2012 and 2013. A steep drop was observed in 2014, with only seven components included. Between 2015 and 2017, eleven components were included in the PP.

The four prevention components 'any form of social worker counselling', 'male condoms', 'awareness raising materials' and 'HIV testing and counselling' were the most commonly delivered components in the Ukrainian PP across all years. Lubricants, 'any form of STI testing' and 'referrals to relevant specialists' were also consistently included in the prevention package in all years, except for 2014. 'Overdose prevention' was only included once in 2012, while 'online counselling', 'training about safe behaviour', 'case management', 'humanitarian aid'

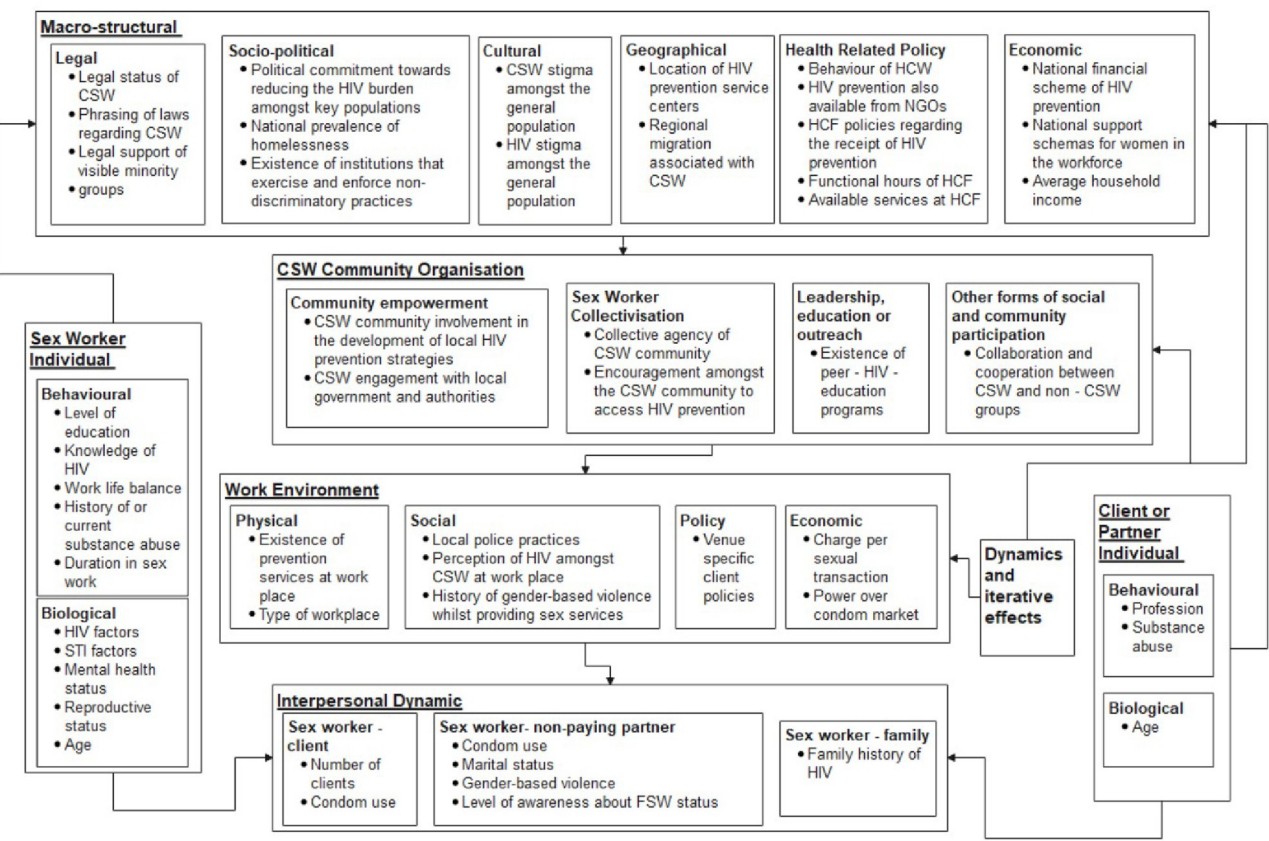

**Fig 2. Conceptual framework of factors and structural levels influencing access to HIV prevention.**

and 'tea and meals' were only included twice. The level of coverage of each component may vary based on region.

## Results of exploratory analysis of IBBS

26.79% of FSWs included in the overall study sample were recruited in the West of Ukraine, 20.06% in the South, 22.74% in the East and 30.41% in Central Ukraine. The average age was 28.52 years, 27.58% of the sample was living with a partner and 23.63% had obtained tertiary education (Table 3). In 2015, 99.12% of participants were female and 0.88% were male. In 2017, 97.72% of participants were female, 2.14% male and 0.13% transgender. No data about gender was available for 2009, 2011 and 2013.

**Prevention package coverage analysis (objective 3).** Fig 3 represents the coverage of both the basic and extended definitions of the PP amongst Ukrainian FSW from 2009 to 2017. For both definitions of access to prevention, coverage increased from 2009, peaked in 2013 and then decreased in 2015 and 2017. The highest proportion of FSWs who had access to the PP was in 2013, where 51.46% and 67.42% accessed the basic and extended PP, respectively, whereas the lowest level of coverage was seen in 2017 with only 26.71% received the basic and 32.10% the extended PP, respectively. Between 48.4% and 54.09% of Ukrainian FSW received HIV testing between 2009 and 2015, however, this drastically decreased in 2017 with only 27.92% coverage. Condom coverage presented a similar trend, as between 52.85% and 65.36% of FSW received condoms from 2009 to 2015, yet only 46.72% in 2017.

**Table 2. Overview of components of the PP provided by Alliance from 2008 to 2017.**

| Component \ Year | 2008 | 2009 | 2010 | 2011 | 2012 | 2013 | 2014 | 2015 | 2016 | 2017 | Sum |
|---|---|---|---|---|---|---|---|---|---|---|---|
| Any form of counselling from a social worker | ■ | ■ | ■ | ■ | ■ | ■ | ■ | ■ | ■ | ■ | 10 |
| Male condoms | ■ | ■ | ■ | ■ | ■ | ■ | ■ | ■ | ■ | ■ | 10 |
| Awareness raising and educational material | ■ | ■ | ■ | ■ | ■ | ■ | ■ | ■ | ■ | ■ | 10 |
| Any form of HIV testing and counselling | ■ | ■ | ■ | ■ | ■ | ■ | ■ | ■ | ■ | ■ | 10 |
| Lubricants | ■ | ■ | ■ | ■ | ■ | ■ |  | ■ | ■ | ■ | 9 |
| Any form of STI testing | ■ | ■ | ■ | ■ | ■ | ■ |  | ■ | ■ | ■ | 9 |
| Referrals to relevant specialists | ■ | ■ | ■ | ■ | ■ | ■ |  | ■ | ■ | ■ | 9 |
| Some form of peer counselling |  | ■ | ■ | ■ | ■ | ■ |  | ■ | ■ | ■ | 8 |
| Any form of specialist medical or legal counselling | ■ | ■ | ■ | ■ | ■ | ■ |  |  |  |  | 6 |
| Any form of hepatitis B testing |  |  |  | ■ | ■ | ■ |  | ■ | ■ | ■ | 6 |
| Antiseptics |  | ■ | ■ | ■ | ■ | ■ | ■ |  |  |  | 6 |
| Pregnancy tests |  | ■ | ■ | ■ | ■ | ■ | ■ |  |  |  | 6 |
| Organisation of leisure activities |  | ■ | ■ | ■ | ■ | ■ |  |  |  |  | 5 |
| Childcare and services |  | ■ | ■ | ■ | ■ | ■ |  |  |  |  | 5 |
| Cosmetologist and hairdresser services |  | ■ | ■ | ■ | ■ | ■ |  |  |  |  | 5 |
| Any form of professional/ skilled work training |  | ■ | ■ | ■ | ■ | ■ |  |  |  |  | 5 |
| Participation in group work |  | ■ | ■ | ■ |  | ■ |  |  |  |  | 4 |
| Female condoms and training for usage |  |  | ■ | ■ | ■ | ■ |  |  |  |  | 4 |
| Some form of TB screening |  |  |  | ■ |  |  |  | ■ | ■ | ■ | 4 |
| Counteracting violence |  |  |  | ■ | ■ | ■ | ■ |  |  |  | 4 |
| Any form of hepatitis C testing |  |  |  | ■ | ■ | ■ |  |  |  |  | 3 |
| Referral to opioid substitution therapy (OST) |  |  |  |  |  |  |  | ■ | ■ | ■ | 3 |
| Syringe exchange and/or delivery |  | ■ | ■ |  | ■ |  |  |  |  |  | 3 |
| Distribution of general medication |  |  |  | ■ | ■ | ■ |  |  |  |  | 3 |
| Basic household services |  | ■ | ■ | ■ |  |  |  |  |  |  | 3 |
| Intimate hygiene items | ■ | ■ | ■ |  |  |  |  |  |  |  | 3 |
| Online counselling |  |  |  |  | ■ | ■ |  |  |  |  | 2 |
| Humanitarian aid |  | ■ | ■ |  |  |  |  |  |  |  | 2 |
| Tea and meals |  | ■ | ■ |  |  |  |  |  |  |  | 2 |
| Training about safe behaviour | ■ |  |  |  |  | ■ |  |  |  |  | 2 |
| Case Management |  |  |  |  | ■ | ■ |  |  |  |  | 2 |
| Overdose prevention |  |  |  |  | ■ |  |  |  |  |  | 1 |
| Total components per year | 10 | 21 | 22 | 23 | 24 | 24 | 7 | 11 | 11 | 11 |  |

**Descriptive longitudinal group comparison (objective 4).**   When stratified by receipt and non-receipt of the basic and extended PP, our descriptive longitudinal group comparison found only marginal differences and therefore, only results for the basic PP are shown and discussed. Results for the extended PP are available in S5 Table. Additionally, due to the volume of data, only results from the descriptive longitudinal group comparison that (i) showed a clear difference between those that did access the basic PP compared to those that did not or (ii) were frequently discussed in the literature as being critical, are explicitly described in our results. For some thematic clusters, no suitable variables were found in the IBBS datasets; for these clusters, we could not conduct further analysis.

In relation to the factors that operate on a *macrostructural level*, our descriptive comparison produced mixed results. Amongst the individual-level proxies that we used for representing macrostructural relationships, the belief that HIV testing was not personally available, was less

**Table 3. Demographic characteristics of participants in IBBS.**

| | Total *N (%)* | 2009 *N (%)* | 2011 *N (%)* | 2013 *N (%)* | 2015 *N (%)* | 2017 *N (%)* |
|---|---|---|---|---|---|---|
| **Participants** | - | N = 2278 | N = 5023 | N = 4806 | N = 4300 | N = 5043 |
| **Mean age (range)** | 28.52 (14–62) | 27.38 (14–55) | 27.65 (14–53) | 28.51 (14–62) | 28.87 (15–61) | 29.63 (15–61) |
| *Gender* | | | | | | |
| **Female** | N/A | - | - | - | 99.12 | 4928 (97.72) |
| **Male** | N/A | - | - | - | 0.88 | 108 (2.14) |
| **Transgender** | N/A | - | - | - | - | 7 (0.14) |
| *Region of Ukraine* | | | | | | |
| **Western** | 5747 (26.79) | 703 (30.86) | 1250 (24.89) | 1000 (20.81) | 1200 (27.91) | 1594 (31.61) |
| **Southern** | 4303 (20.06) | 150 (6.58) | 1103 (21.96) | 1350 (28.09) | 800(18.6) | 900 (17.84) |
| **Eastern** | 4877 (22.74) | 462 (20.28) | 1260 (25.08) | 1356 (28.21) | 900 (20.93) | 899 (17.83) |
| **Central** | 6523 (30.41) | 963 (42.27) | 1410 (28.07) | 1100 (22.89) | 1400 (32.56) | 1650 (32.72) |
| *Partnership status* | | | | | | |
| **Living with a partner** | 5915 (27.58) | 749 (32.88) | 1290 (25.68) | 1262 (26.26) | 1332 (30.98) | 1282 (25.42) |
| **Not living with partner** | 15535 (72.42) | 1529 (67.12) | 3733 (74.32) | 3544 (73.74) | 2968 (69.02) | 3761 (74.58) |
| *Education* | | | | | | |
| **< Tertiary education** | 16362 (76.37) | 1805 (79.34) | 3881 (77.26) | 3856 (80.23) | 3175 (74.06) | 3645 (72.41) |
| **≥Tertiary education** (attended vocational training OR higher form of education) | 5063 (23.63) | 470 (20.66) | 1142 (22.73) | 950 (19.76) | 1112 (25.94) | 1389 (27.59) |

common amongst PP recipients compared to PP non-recipients in all years. In all years except 2009, of the FSWs who believed that HIV testing was not personally available, fear of HIV status being exposed and no desire to know HIV test results, two individual-level proxies addressing the level of HIV stigma amongst the population, were less common amongst the PP recipients. In all years except 2011, belief that HIV prevention is too expensive, the individual-level proxy referring to the national financial scheme for HIV prevention, was less common amongst the PP recipients compared to the PP non-recipients. In 2009 and 2011, dissatisfaction with HCF staff attitude and the opening hours of HCFs, two individual-level proxies addressing health-related policy, were more common amongst PP recipients. In all years, sex work as the sole form of income, the individual-level proxy used to address support schemas for women (re)-entering the workforce, was consistently more common amongst PP recipients than PP non-recipients.

Factors that operate on both the *community organisational level* and the *work environment level* identified consistent differences across all years between PP recipients and PP non-recipients. The proportion of FSWs, who were clients of an NGO was much larger in PP recipients (83–97%) than in PP non-recipients (15–25%). Across all years, soliciting on the street and on highways was more common amongst PP recipients, whereas soliciting in affluent settings was less common. Of those who experienced GBV, PP recipients were consistently more affected. Of these FSWs, GBV from clients or police officers was more common amongst the PP recipients than the PP non-recipients.

Regarding factors that operate on the *interpersonal dynamic level*, having sex without a condom was less common, while having a partner that knew of their FSW status was more common amongst PP recipients compared to PP non-recipients in all years.

Analyses of the factors that operate on a *sex worker individual level* showed that sharing needles was more common amongst PP recipients, except in 2011. PP recipients were in 2009,

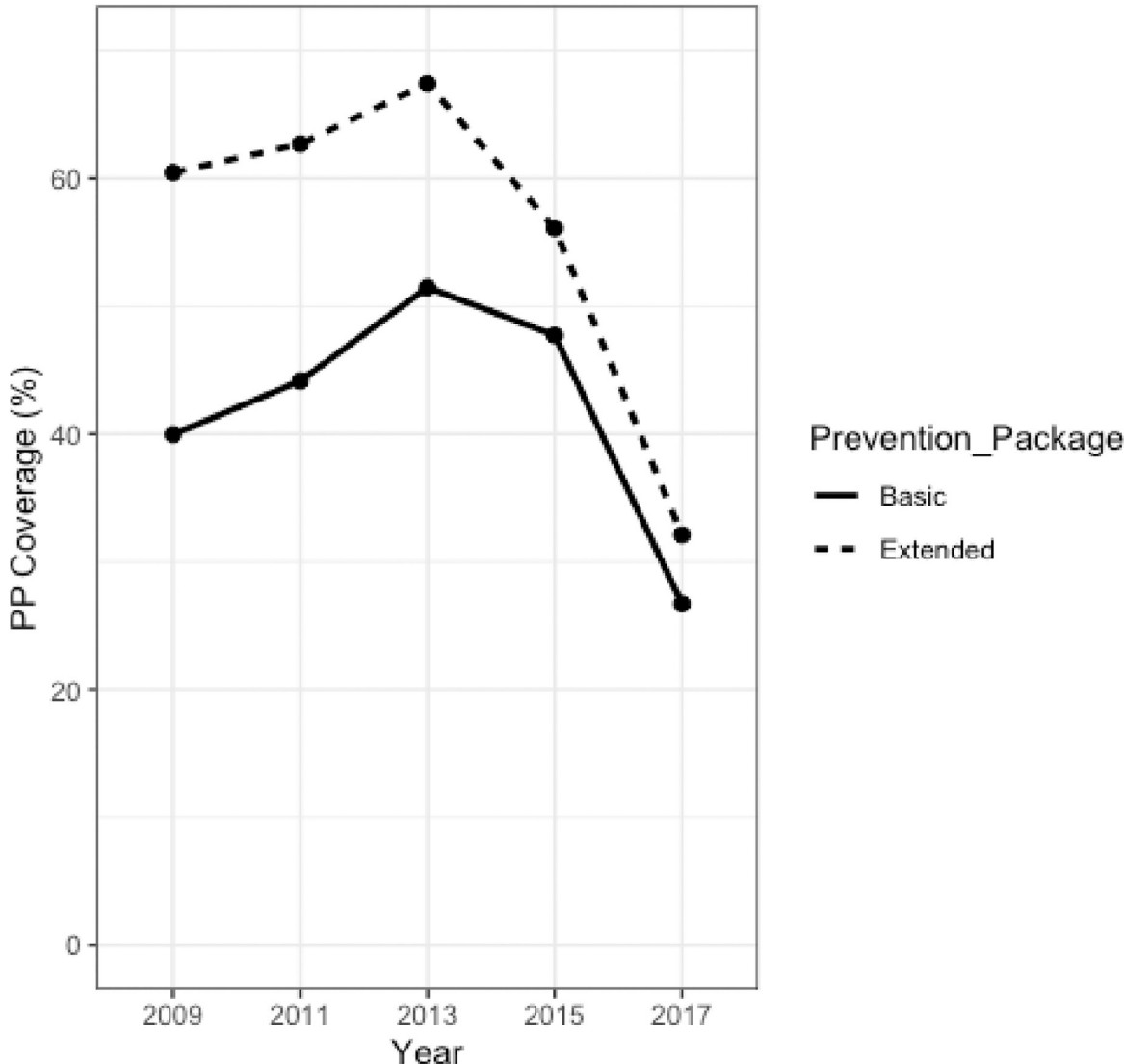

**Fig 3. Coverage of the basic and extended PP in FSW in the Ukraine between 2009 and 2017.**

2011 and 2013 on average one year–and in 2015 and 2017, two years older than their counterparts. Both groups increased in age over time, from 28 to 32 and 27 to 29 years amongst PP recipients and PP non-recipients, respectively. Knowledge of HIV transmission routes and knowledge of where to access prevention was consistently higher amongst PP recipients compared to non-recipients.

## Discussion

In this study, we identified factors that acted as potential barriers and facilitators to the access of HIV prevention amongst the Ukrainian FSW population between 2009 and 2017. Our literature-informed conceptual framework identified various risk factor clusters, which operate on six structural levels. Our exploratory analyses identified that these factors may have partially been responsible for the changes in PP coverage, which peaked in 2013 and then dropped in

both 2015 and 2017. Our results suggest that sex work as the only form of income, being a client of an NGO, street and highway solicitation, partner's knowledge of occupation, knowledge of where to access prevention and knowledge of HIV transmission routes may act as important facilitators to HIV prevention. Potential barriers to accessing the PP were soliciting in affluent settings and unsafe sex.

The initial conceptual framework reflected the findings from empirical studies, which were conducted in the Americas, the African, East Mediterranean, European, South-East Asia and West Pacific regions between 2005 and 2018. To tailor this global conceptual framework to a Ukrainian setting, we discussed our findings with staff members in the PHC of the Ministry of Health of Ukraine and subsequently removed one determinant–sex worker family–from the framework, as it was not applicable to the Ukrainian context. The resulting conceptual framework, thus, reflects both current, global literature as well as local insights into barriers and facilitators to accessing PP in Ukraine. This conceptual framework can be utilised to inform future research on barriers and facilitators of accessing HIV prevention for FSWs and contribute to advancing theories underlying this field of research.

Findings from our document analysis identified that the Ukrainian PP has changed over the past nine years, which may be attributed to the fluctuation in international funding and the armed conflict in the east of Ukraine. Between 2008 and 2013, Alliance's international funding grew from US$28.4 to US$37.7 million, which coincided with an expansion in the PP from 10 to 24 components in 2008 and 2013, respectively [27, 32]. Only seven components were offered in 2014, the year in which Ukraine experienced a severe restructuring of the political landscape [44], recession of the economy [45] and the onset of the continuing armed-conflict in the east [46]. As one of the only NGOs permitted to work in the conflict zones of the east, Alliance diverted a large proportion of international funding to the emergency procurement of HIV and tuberculosis (TB) medicine in these eastern regions [32, 33]. This diversion of funds may be one factor to explain the sharp decrease in PP components in regions of Ukraine from 2014 onwards, particularly, specialist medical or legal counselling. Despite this cutback, the components that were included from 2015 to 2017 did largely meet the WHO recommendations, which included condoms, HIV testing and counselling, consultations with social workers as well as screening and prevention of co-infections such as TB and Hepatitis B [47].

The coverage of both the basic and the extended PP also changed over this nine-year period. Between 2009 and 2013, progress toward the "five pillars for achieving less than 500,000 new infections by 2020", a sub-target of the 90-90-90 goals seemed promising as the level of extended PP coverage amongst Ukrainian FSWs increased from 60% to 67%. However, although Kyiv and other cities signed the Paris Declaration on Fast-Track Cities in 2014, a pact that committed cities to focusing on vulnerable and marginalized people, including key populations, [22] extended PP coverage dropped to 32% in 2017. Coincidentally, Alliance introduced a new policy in 2017 that deemed FSWs who knew of their HIV positive status ineligible for testing [48]. This may explain the steep decline in both HIV testing coverage and subsequently, the sharp drop in PP coverage in 2017. Nonetheless, our analyses also identified a 18.05% decrease in condom coverage in 2017, hence, future research is required to further investigate this trend. Alliance defines receipt of basic PP as 'the receipt of at least one male condom and at least one consultation with a social worker', however, due to limitations in the IBBS questionnaires, our definition included, 'receipt of male condoms and receipt of HIV testing and counselling'. This could explain the discrepancy between the coverage we observed, and that reported by Alliance. Alliance report that the basic PP coverage remained relatively stable over time (46.7% in 2013 and 48% in 2017), whereas we identified a clear decrease in basic PP coverage over this time period (51.46% and 26.71% in 2013 and 2017, respectively).

Furthermore, Alliance utilises an internal database, Syrex for reporting [32] as opposed to the IBBS, which may also explain this difference in coverage.

The fall in coverage between 2013 and 2017 may further be explained by the reduction in international financial support and the redistribution of resources associated with emergency programmes in the conflict zone, both of which operate on a *macrostructural level*.

Changes in the PP coverage may also be explained by specific factors associated with access to prevention. Our findings from the *macrostructural level* suggest that relying upon sex work as one's sole source of income consistently acted as a facilitator to the basic PP. We identified that the proportion of FSWs relying upon sex work as their only form of income increased in both groups between 2015 and 2017, which coincides with and could potentially have arisen because of the outbreak of the armed conflict in the East of Ukraine. This spike in reliance upon sex work aligns with previous studies that outline the catalytic effect that armed conflict can have upon internal displacement and economic insecurity, which in turn, often results in conditions that facilitate engagement in sex work as a source of income, particularly amongst woman [49–51].

We observed a high level of stigma (54%) amongst PP recipients who feared their HIV status being exposed in 2009, however we did not observe meaningful between-group differences. Previous studies have deemed stigma to be one of the most salient barriers to HIV prevention [17, 52, 53], fuelled by other factors on different structural levels such as criminalisation [15, 54–56] on the *macrostructural level* and a lack of community empowerment on the *community organisational level* [17, 57, 58].

Our results identified an encouraging trend underscoring the importance of facilitators that operate on a *community organisation level*. In congruence with previous literature [17, 57–59], our findings strongly imply that collective agency of the Ukrainian FSW community via NGO membership largely facilitated access to the basic PP, as most PP recipients were clients of an NGO. Knowledge of HIV transmission routes was higher amongst PP recipients across all years, which may be attributable to consultations with social workers and the distribution of HIV awareness-raising materials, two components that were consistently included in the PP. Authors have also identified that social and community participation can foster transparency and trust between FSWs and non-FSWs, which can increase their access to prevention [39, 60]. Our results support this in all years, as disclosing one's FSW status to a partner was more common amongst PP recipients, suggesting that this may have acted as a facilitator.

Our results from the *interpersonal dynamic* and the *work environment level* not only elucidate potential barriers and facilitators to PP in the Ukrainian context but highlight the interplay of factors that operate on different structural levels. We found condom use to be consistently more common amongst PP recipients than PP non-recipients, this may suggest that sex without a condom could act as a barrier to the PP. An array of studies have identified that unsafe sex is associated with GBV, a factor operating on both the *work environment* and the *interpersonal dynamic level*. Authors have described that FSWs who have experienced GBV in the work environment are less likely to instigate condom negotiation in fear of client aggression [61–66] and therefore, may be less inclined to seek condoms, an important component of the Ukrainian PP. We identified an alarmingly high prevalence of GBV amongst both groups at the *work environment*, however, contrary to the literature, GBV at the *work environment level* was more common amongst PP recipients than PP non-recipients. Results measuring GBV at the *interpersonal level* were not meaningful. It is possible that the criminalisation of sex work, a barrier that operates on the *macrostructural level*, may have led to an increase in GBV [65, 67, 68] in the workplace, and in turn, an increase in unsafe sex [69]. Although recent literature supports this hypothesis [15, 55, 70], future research is required to further explore this

within the Ukrainian context. Further research is particularly warranted to investigate the higher prevalence of GBV amongst PP recipients and this association with condom use.

Two additional factors on the *work environment level*, street- and highway-based soliciting appeared to act as facilitators to the basic PP in all years. This contrasted with previous literature, which has highlighted that street-based FSWs are less likely to access HIV prevention due to an array of barriers, such as higher mobility, higher risk-taking behaviour and higher exposure to GBV [9, 61, 71, 72]. The facilitating effect that street- and highway soliciting had upon access to the PP in the Ukrainian context may be attributed to the successful employment of mobile clinics, an intervention conducted by Alliance to ease access to HIV prevention services, specifically targeting these vulnerable subgroups [34]. Affluent-based FSWs may be less likely to disclose their HIV status due to fear of being associated with HIV and being banned from a venue, resulting in loss of income [2, 21, 39, 73]. Furthermore, affluent-based FSWs may be less interested in accessing the basic PP, as they are able to afford these services. This warrants future qualitative research to investigate the prevention needs of Ukrainian FSWs, relative to their soliciting setting.

Our findings from the *sex worker individual level* corroborate previous studies, which have demonstrated that knowledge of where to access prevention acts as a key facilitator [9, 39, 56, 57, 59, 74]. However, contrary to our expectations, we found that more than 82% of Ukrainian FSWs who did *not* access the basic PP *knew* where to access prevention. This may suggest the need to further address macrostructural barriers in efforts to ameliorate access to HIV prevention.

Although our study provides valuable insights into the factors associated with access HIV prevention, future research is needed to deepen this knowledge. Our analyses were limited by the quality of the data and further quantitative analyses, both in Ukrainian but also in other specific and broader contexts, should focus on collecting high-quality and valid data to address both specific associations, as well as more complex, multi-component associations. Qualitative research could provide evidence on opinions, perceptions and motivations of FSWs, which may help to gain a deeper insight into the source of barriers such as stigma, GBV and the influence of the armed conflict upon access to HIV in a Ukrainian context. Future qualitative studies exploring the specific needs of FSWs, namely, which PP components are most necessary, may also elucidate important findings for future prevention programmes.

## Strengths and limitations

The strength of this work lies in its theoretical foundation in current empirical literature and the use of a conceptual framework to inform the analyses. Using a conceptual framework is not only considered a means of conceptualizing and structuring research, it also contributes to theory building around the topic of interest. Our conceptual framework informed the selection of variables in the survey, hence contributing to the structure of the exploratory analyses. Additionally, the use of multiple, large datasets from FSWs in almost all regions of Ukraine allowed us to explore changes over time, another major strength of this study.

Several limitations need to be taken into account when interpreting these results. Firstly, we limited our literature searches to one database, PubMed. Our search strategy did not focus on identifying conceptual papers that presented and discussed theories, models or frameworks of barriers and facilitators but empirical papers assessing barriers and facilitators in real-world settings. There were also limitations with regard to the assessed IBBS surveys. Changes in the design of surveys made it difficult to compare particular factors across time, particularly questions measuring barriers on the *macrostructural level*. Additionally, missing data prevented us from measuring informative outcomes, such as the coverage of the basic PP, as defined by Alliance. The nature of the questions themselves may have also influenced results; both FSW and

HIV are two highly sensitive issues and therefore, recall and social desirability bias may have impacted the responses in the survey, particularly since the survey were administered face-to-face. There were also limitations in our analyses. In measuring PP coverage, changes in the specific components of the PP over time resulted in our indicator for extended PP coverage being more volatile and less informative than the basic PP. However, a similar trend was seen in both basic and extended PP coverage, hence this did not substantially impact the measurement of coverage. Nonetheless, the level of coverage of each PP component may vary based on region and therefore have had some influence upon our results. Geographical sampling of cities included in the IBBS may have also influenced the level of coverage. As in other studies that utilised IBBS data [73, 75–77], we did not weight for sampling and therefore, our results are to be interpreted with this in account. Only in 2017 and 2015 was a question on gender included in the IBBS and although only a small proportion of SWs were male or transgender, limitations caused by the misclassification need to be considered.

We conducted and reported descriptive statistical analyses, however, due to limitations in the data, such as inconsistent response scales and a lack of internal consistency throughout all five datasets alongside structural changes between 2013 and 2015. Hence, the analyses should be considered exploratory rather than confirmatory and should be interpreted cautiously. As the IBBS measured individual behaviours, we were limited to assigning individual-level proxies analyse factors that influence access to the Ukrainian PP on the macrostructural level. RDS and TLS have found to be effective sampling methods for reaching marginalised populations. In accordance with others working with the IBBS data [2, 78], we did not consider survey weights when calculating descriptive statistics, which may have led to an over-representation of certain subpopulations.

## Conclusion

To our knowledge, this is the first study to explore the factors associated with the access of Ukrainian FSW to HIV prevention on a national scale across a time span of nine years. Our study strengthens current knowledge that access to prevention is complex with many factors found on multiple levels and that access to prevention has decreased since 2014. We also identified several factors such as relying solely upon sex work for income, being a client of an NGO, knowledge of HIV, partner's knowledge of occupation, soliciting setting and condom use that may be associated with access to HIV prevention within the Ukrainian context. Future interventions targeting these factors may increase the coverage of HIV prevention and thus, contribute to reducing the HIV burden within the FSW population.

## Supporting information

**S1 Appendix. Literature review method, protocol and results.**
(DOCX)

**S1 Table. Formation of thematic clusters used to populate the conceptual framework.**
(DOCX)

**S2 Table. Categories and specific PP components as extracted from the Alliance general reports from 2008 to 2017.**
(DOCX)

**S3 Table. PP component and corresponding IBBS item for 2009, 2011, 2013, 2015 and 2017.**
(DOCX)

**S4 Table. Independent variables from IBBS and corresponding thematic clusters for 2009, 2011, 2013, 2015 and 2017.**
(DOCX)

**S5 Table. Barriers and facilitators to accessing HIV prevention stratified by receipt or non-receipt of extended PP between 2009 and 2017.**
(DOCX)

## Acknowledgments

The paper summarizes and analyses the data from studies funded by of the Global Fund to Fight AIDS, Tuberculosis and Malaria (TGF) and implemented in Ukraine by the International Charitable Foundation "Alliance for Public Health" (Alliance). The data for analysis were provided to the authors by the data holder the Public Health Centre (PHC) of the Ukrainian Ministry of Health. All opinions presented in this manuscript belong to the authors alone and not represent position of the Global Fund to Fight AIDS, Tuberculosis and Malaria, Alliance for Public Health or the Public Health Centre (PHC) of the Ukrainian Ministry of Health. We would like to thank Ihor Kuzin (PHC), Yana Sazonova (Alliance), Roksolana Kulchynska (CDC) and Tetyana Salyuk (Alliance) for their comments on the manuscript, Oleksandr Korotych (PHC) and Oksana Kovtun (Alliance) for providing us with the IBBS datasets and, Nadiya Yanhol (Alliance) for her insight into the PP. Lastly and most importantly, thank you to the SWs that partook in the IBBS studies.

## Author Contributions

**Conceptualization:** Natasha Blumer, Lisa M. Pfadenhauer, Jacob Burns.

**Formal analysis:** Natasha Blumer, Jacob Burns.

**Methodology:** Natasha Blumer, Lisa M. Pfadenhauer, Jacob Burns.

**Supervision:** Lisa M. Pfadenhauer, Jacob Burns.

**Visualization:** Natasha Blumer.

**Writing – original draft:** Natasha Blumer.

**Writing – review & editing:** Natasha Blumer, Lisa M. Pfadenhauer, Jacob Burns.

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
