## [Decision Letter · Decision Letter 0]

3 Nov 2020

PONE-D-20-24729

Access to HIV-prevention in female sex workers in Ukraine between 2009 and 2017: coverage, barriers and facilitators

PLOS ONE

Dear Dr. Blumer,

Thank you for submitting your manuscript to PLOS ONE. After careful consideration, we feel that it has merit but does not fully meet PLOS ONE’s publication criteria as it currently stands. Therefore, we invite you to submit a revised version of the manuscript that addresses the points raised during the review process.

We look forward to receiving your revised manuscript.

Kind regards,

Zixin Wang, PhD.

Academic Editor

PLOS ONE

Journal Requirements:

"Data collection for this study was undertaken while NB was affiliated to PHC. I have read the journal's policy and the authors of this manuscript have the following competing interests: Data collection for this study was undertaken while NB was affiliated to PHC. All opinions presented in this manuscript belong to the author alone, and not any institution to which they are or were affiliated. The authors declare that they have no competing interests."

i) Please confirm that this does not alter your adherence to all PLOS ONE policies on sharing data and materials, by including the following statement: "This does not alter our adherence to  PLOS ONE policies on sharing data and materials.” (as detailed online in our guide for authors http://journals.plos.org/plosone/s/competing-interests).  If there are restrictions on sharing of data and/or materials, please state these. Please note that we cannot proceed with consideration of your article until this information has been declared.

ii) Please include your updated Competing Interests statement in your cover letter; we will change the online submission form on your behalf.

Reviewers' comments:

Reviewer's Responses to Questions

**Comments to the Author**

1. Is the manuscript technically sound, and do the data support the conclusions?

Reviewer #1: Partly

Reviewer #2: Yes

Reviewer #3: Yes

2. Has the statistical analysis been performed appropriately and rigorously? 

Reviewer #1: No

Reviewer #2: Yes

Reviewer #3: No

3. Have the authors made all data underlying the findings in their manuscript fully available?

Reviewer #1: No

Reviewer #2: Yes

Reviewer #3: Yes

4. Is the manuscript presented in an intelligible fashion and written in standard English?

Reviewer #1: Yes

Reviewer #2: Yes

Reviewer #3: No

5. Review Comments to the Author

Reviewer #1: In this study, the authors carried out a comprehensive literature search, a document analysis, and a quantitative analysis of survey data collected between 2009 and 2017 to identify factors associated with the access of HIV prevention in Ukraine sex workers. The topic is significant, and the description of methodological details are sufficient. However, I find that the current form of the manuscript is rather lengthy. Specifically, it is difficult to understand how the document search of the Ukrainian prevention package relates to the other objectives. Only very basic bivariate-level descriptive analysis was used to analyze the survey data. Please see below for my specific comments.

Major comments:

1. Four objectives were stated in this study, which can be distracting. I find the essential question of this study is to assess the coverage of PP in Ukraine over time and understand factors associated with the access. The development of a conceptual framework, rather than presenting it as a separate objective, can be considered as the necessary step of literature review for identifying putative factors of interest.

2. How is the document search relates to the analysis of the questionnaire data? I think it can be more clearly explained.

3. The authors repeatedly used “factors that influence” access to PP throughout the manuscript. However, limited by the current cross-sectional design and bivariate analysis, I think such wording infers a causal relationship and should be revised to “factors associated”.

4. About the sampling design, it seems that the surveys are conducted independently of each other. Is it possible that one respondent was selected multiple times? Any identifier that can explore this issue?

5. Regarding the survey question design, to what extent are the questions comparable? If the questions are designed in similar ways and there are ways to uniquely identify the participants, one way to improve the analysis is to pool the data together, and include calendar year as an independent variable to test the effect of calendar year. If one individual is sampled multiple times, multilevel analysis can be considered by treating the individual as a level two variable. The study in its current form is a rather scattered analysis of survey data conducted in different years. Given that these surveys essentially adopted a non-probability sampling approach, any conclusions regarding trends or trajectory are weak.

6. In the variable selection section, only the outcome variables were described. Independent variables should also be systematically presented in the Method section.

7. To understand factors associated with access, rather than presenting a big table of descriptive analysis by group, the authors may consider using multiple logistic regressions. The outcome variable should be whether access to PP, while the independent variables should be a selection of variables investigated in Table 3.

8. If I understand correctly, measures of the macrostructural factors are all obtained from individual-level response. They are not upper-level objective measures and are subject to measurement bias. In general, these independent variables need to be much more carefully presented and discussed.

9. Did the authors obtain ethical approvals as human subjects were involved?

Minor comments:

1. Objectives do not need to be presented as a separate section.

2. In the Introduction, the WHO guideline was published in 2014. The authors then stated that in 2006, the Alliance launched a PP in Ukraine adhering to WHO guidelines. The timeline is a bit confusing. Better to clarify.

3. What does it mean by “no missing data were excluded”. Are there missing values?

Reviewer #2: The authors of this manuscript managed to present the overall picture of access to HIV-prevention among female sex workers in Ukraine over nearly a decade. After reading the manuscript, it is obvious that the authors have done extensive work and put in many efforts to explore the barriers and facilitators of access to prevention service in key populations. The manuscript is clearly structured and well-written, supported by literature, data and other materials indicating how the author performed the literature review and exploratory analysis. There are only several questions I would like the author to give a more detailed explanation.

1. In method, the author mentioned that the definition of extended PP receipt was adapted by increasing the timeframe. Is the any modification in the number of service received given the timeframe was longer in the new definition? The definition of extended PP receipt was not clearly stated either in the main document or in the supporting material. As far as I am concerned, even the package was different year from year, the number of service received should stay consistent, just like the UNAIDS definition (i.e. two service received in three months)

2. In method, the author suggested no missing data were excluded. Judging from the data, numbers of missing values were not indicated in the table. It would be great if the author could give further explanation about how the missing data were processed.

3. In results, the author compared characteristics between recipients and non-recipients, but there was no statistical testing. If possible, it could be great if the author could provide more statistical support for between group comparison.

4. In discussion, the authors mentioned the armed conflicts in the eastern Ukraine might affect the service coverage. It would be great if the author could brief us about the location of sampling cities in method so that the audience could have a clearer understanding about the extent of the impact the conflicts had on the service coverage.

5. In discussion, the authors suggested GBV was more common in PP recipients than non-recipients. Yet, in the table, only data of the year 2013 and 2015 support this claim. It would be better if the author could provide data to better explain this statement.

6. Besides, the authors showed that consistent condom use and GBV were both more prevalent among PP recipients, meanwhile suggested FSW experienced GBV may be less inclined to seek condoms. These two facts contradicted with each other. It could be great if the author could settle this contradiction that why PP recipient which experienced more GBV than non-recipients displayed more consistent condom use behavior.

7. Given the fact that the current design and analysis were inadequate for a causal reference, the statement in discussion: “sex without a condom may impede access to the PP” sounds a little inappropriate.

8. Overall, this manuscript utilized longitudinal data from a series of national survey with about 5 waves. However, the “trend” of service coverage in this paper was not supported by testing for trends. It would be great if the author could provide more statistical evidence to support the description of coverage changes over the years.

Reviewer #3: Comments:

This paper has comprehensively described the access to HIV-prevention in female sex workers in Ukraine between 2009 and 2017: coverage, barriers and facilitators. While this article has provided numerous information including literature review, document analysis, exploratory analysis. The rich information is not easy to follow to generate a whole story to a specific research question. It has four objectives, which in my opinion is way too long for an article. The contents look more like a project report instead of one original research. I would suggest the authors to shorten the article and specify research questions instead of lump everything together and resubmit again.

Line 82, the reference is not right.

6. PLOS authors have the option to publish the peer review history of their article (what does this mean?). If published, this will include your full peer review and any attached files.

Reviewer #1: No

Reviewer #2: No

Reviewer #3: No

---

## [Author Response · Author response to Decision Letter 0]

8 Feb 2021

COMMENT REVIEWER 1

1. In this study, the authors carried out a comprehensive literature search, a document analysis, and a quantitative analysis of survey data collected between 2009 and 2017 to identify factors associated with the access of HIV prevention in Ukraine sex workers. The topic is significant, and the description of methodological details are sufficient. 

2. However, I find that the current form of the manuscript is rather lengthy. Specifically, it is difficult to understand how the document search of the Ukrainian prevention package relates to the other objectives. Only very basic bivariate-level descriptive analysis was used to analyze the survey data. Please see below for my specific comments.

CHANGE MADE

No changes made.

JUSTIFICATION

1. We thank the reviewer for their positive feedback.

2. Please, see the corresponding answer to each comment below. 

MAJOR COMMENT REVIEWER 1

Four objectives were stated in this study, which can be distracting. I find the essential question of this study is to assess the coverage of PP in Ukraine over time and understand factors associated with the access. The development of a conceptual framework, rather than presenting it as a separate objective, can be considered as the necessary step of literature review for identifying putative factors of interest.

CHANGE MADE

Please, see revisions in the Conceptual Framework section of the Method, the Conceptual Framework (objective 1) section of the Results, and the Supporting Information Captions. We also created a new supporting information document, S1 Appendix. Literature review method, protocol and results.

JUSTIFICATION

We thank the reviewer for their comments. We have removed the narrative details about the literature review to both reduce the length of the manuscript and focus upon the importance of the conceptual framework in the Method. All information is available in the Supporting Information, should the reader wish to scrutinise details about the literature review, see S1 Appendix. Literature review method, protocol and results.

We do, however, believe that a conceptual framework should be presented as a separate objective in the Methods. To our knowledge, there is no existing framework that presents the current evidence-based factors associated with access to HIV prevention amongst female sex workers (FSW) and highlights which structural level these factors operate on. Only two studies have similar aims; however, we believe our framework is somewhat more comprehensive and thus worthy of a designated section. The systematic review conducted by Nnko and colleagues only targets FSWs in sub-Saharan Africa and only included HIV testing and counselling. Whilst the framework Shannon et al (2015) largely informed our research and provided the basic structure of our framework, our framework presents the specific factors associated with access to HIV prevention. 

MAJOR COMMENT REVIEWER 1

How is the document search relates to the analysis of the questionnaire data? I think it can be more clearly explained.

CHANGE MADE

Please, see changes in the first paragraph, the first sentence of the Review of IBBS questionnaires and variable selection and the first sentence of the Outcome Variable section in the Method.

JUSTIFICATION

We have rephrased the text to clarify how the document analysis relates to the questionnaire data in the Method.

MAJOR COMMENT REVIEWER 1

The authors repeatedly used “factors that influence” access to PP throughout the manuscript. However, limited by the current cross-sectional design and bivariate analysis, I think such wording infers a causal relationship and should be revised to “factors associated”.

CHANGE MADE

Please, see changes in the Introduction, Discussion and Conclusions.

JUSTIFICATION

We have made the suggested revisions. 

MAJOR COMMENT REVIEWER 1

About the sampling design, it seems that the surveys are conducted independently of each other. Is it possible that one respondent was selected multiple times? Any identifier that can explore this issue?

CHANGES MADE

No changes made.

JUSTIFICATION

This is indeed an interesting point; there is no unique participant identifier across surveys that would allow this to be tracked.

MAJOR COMMENT REVIEWER 1

Regarding the survey question design, to what extent are the questions comparable? If the questions are designed in similar ways and there are ways to uniquely identify the participants, one way to improve the analysis is to pool the data together, and include calendar year as an independent variable to test the effect of calendar year. If one individual is sampled multiple times, multilevel analysis can be considered by treating the individual as a level two variable. The study in its current form is a rather scattered analysis of survey data conducted in different years. Given that these surveys essentially adopted a non-probability sampling approach, any conclusions regarding trends or trajectory are weak.

CHANGES MADE

No changes made. 

JUSTIFICATION

We appreciate the reviewer's suggestions here on how to improve the analysis.

In principle, we are open to conducting further statistical analyses, however our preference would be to not do so for two reasons: 1) the aim of the study was exploratory in nature, with the potential to suggest aspects for further study (with the appropriate methodology and statistical power) rather than to provide inferential estimates of relationships. Thus, we feel that the descriptive nature of the analyses allows for this exploration, without relying on inferential statistics to determine what is significant and what is not; 2) as described in the Discussion there were some concerns with the underlying data - we feel that the problems with the data do not preclude a rather descriptive and exploratory analysis, however conducting inferential statistics may suggest more confidence than we have in the assessed relationships.

MAJOR COMMENT REVIEWER 1

In the variable selection section, only the outcome variables were described. Independent variables should also be systematically presented in the Method section.

CHANGES MADE

Please see changes in the Independent Variables in the Method section. 

JUSTIFICATION 

We have added a section titled, Independent Variable in the Method section. 

MAJOR COMMENT REVIEWER 1

To understand factors associated with access, rather than presenting a big table of descriptive analysis by group, the authors may consider using multiple logistic regressions. The outcome variable should be whether access to PP, while the independent variables should be a selection of variables investigated in Table 3.

CHANGES MADE

Please see changes made in the Strengths and Limitations section.

JUSTIFICATION

As described above, the aim of the study was exploratory in nature. Given this, we do not feel that further statistical analyses would make the results clearer or more robust. To further emphasise the need for future studies with more robust testing, we have added another sentence into the Strengths and Limitations section. 

MAJOR COMMENT REVIEWER 1

If I understand correctly, measures of the macrostructural factors are all obtained from individual-level response. They are not upper-level objective measures and are subject to measurement bias. In general, these independent variables need to be much more carefully presented and discussed.

CHANGES MADE

Please see changes made in the second paragraph of the Independent Variables in the Method section, the second paragraph of the Descriptive Longitudinal Group Comparison section in the Results, and the last paragraph of the Strengths and Limitations.

JUSTIFICATION

We thank the Reviewer for their comments and have amended the text to emphasise that these variables are individual-level proxies, as no macrostructural level data could be extracted from the IBBS, due to the individual nature of the questionnaire. 

MAJOR COMMENT REVIEWER 1

Did the authors obtain ethical approvals as human subjects were involved?

CHANGES MADE

See changes made in the Review of IBBS questionnaires and variable selection in the Methods section.

JUSTIFICATION

We have inserted a sentence outlining where Alliance received ethical approval, when conducting IBBS. As we conducted a secondary analysis of data, in which ethical approval was already granted, we do not believe it necessary to include that we did not receive ethical approval for this study. 

MINOR COMMENT REVIEWER 1

Objectives do not need to be presented as a separate section.

CHANGE MADE

Please, see the revised Introduction.

JUSTIFICATION

We have made the suggested revisions.

MINOR COMMENT REVIEWER 1

In the Introduction, the WHO guideline was published in 2014. The authors then stated that in 2006, the Alliance launched a PP in Ukraine adhering to WHO guidelines. The timeline is a bit confusing. Better to clarify.

CHANGES MADE

Please see changes made in the second last paragraph of the Introduction.

JUSTIFICATION

We have edited the text to make this sentence clearer. 

MINOR COMMENT REVIEWER 1

What does it mean by “no missing data were excluded”. Are there missing values?

CHANGES MADE

Please, see lines 199-201 of the Method.

JUSTIFICATION

We have amended this sentence and thank the reviewer for bringing it to our attention. 

******

COMMENT REVIEWER 2

The authors of this manuscript managed to present the overall picture of access to HIV-prevention among female sex workers in Ukraine over nearly a decade. After reading the manuscript, it is obvious that the authors have done extensive work and put in many efforts to explore the barriers and facilitators of access to prevention service in key populations. The manuscript is clearly structured and well-written, supported by literature, data and other materials indicating how the author performed the literature review and exploratory analysis. There are only several questions I would like the author to give a more detailed explanation.

CHANGES MADE

No changes made.

JUSTIFICATION

We thank the reviewer for their positive feedback. 

COMMENT REVIEWER 2

In method, the author mentioned that the definition of extended PP receipt was adapted by increasing the timeframe. Is the any modification in the number of service received given the timeframe was longer in the new definition? The definition of extended PP receipt was not clearly stated either in the main document or in the supporting material. As far as I am concerned, even the package was different year from year, the number of service received should stay consistent, just like the UNAIDS definition (i.e. two service received in three months)

CHANGES MADE

Please, see the second paragraph in the Outcome Variable section in the Methods.

JUSTIFICATION

Based on the reviewer's comments we believe that our description of the outcome definition may have been unclear. The only part of the UNAIDS definition that was adapted was the timeframe. This adaptation was solely because most IBBS questions asked about the previous 12 months, rather than the 3 months referenced in the UNAIDS definition. We have added an explicit description of the outcome definition we applied.

COMMENT REVIEWER 2

In method, the author suggested no missing data were excluded. Judging from the data, numbers of missing values were not indicated in the table. It would be great if the author could give further explanation about how the missing data were processed.

CHANGES MADE

Please, see lines 199-201 of the Method.

JUSTIFICATION

We have amended this sentence and thank the reviewer for bringing it to our attention.

COMMENT REVIEWER 2

In results, the author compared characteristics between recipients and non-recipients, but there was no statistical testing. If possible, it could be great if the author could provide more statistical support for between group comparison.

CHANGES MADE

No changes made.

JUSTIFICATION

We appreciate the reviewer's suggestions here on how to improve the analysis.

In principle, we are open to conducting further statistical analyses, however our preference would be to not do so for two reasons: 1) the aim of the study was exploratory in nature, with the potential to suggest aspects for further study (with the appropriate methodology and statistical power) rather than to provide inferential estimates of relationships. Thus, we feel that the descriptive nature of the analyses allows for this exploration, without relying on inferential statistics to determine what is significant and what is not; 2) as described in the Discussion there were some concerns with the underlying data - we feel that the problems with the data do not preclude a rather descriptive and exploratory analysis, however conducting inferential statistics may suggest more confidence than we have in the assessed relationships.

COMMENT REVIEWER 2

In discussion, the authors mentioned the armed conflicts in the eastern Ukraine might affect the service coverage. It would be great if the author could brief us about the location of sampling cities in method so that the audience could have a clearer understanding about the extent of the impact the conflicts had on the service coverage.

CHANGES MADE

Please, see changes made to lines 348-351 in the Discussion.

JUSTIFICATION

We thank the reviewer for their comment and have amended this sentence accordingly. 

COMMENT REVIEWER 2

In discussion, the authors suggested GBV was more common in PP recipients than non-recipients. Yet, in the table, only data of the year 2013 and 2015 support this claim. It would be better if the author could provide data to better explain this statement.

CHANGES MADE

Please, see changes made to the Discussion.

JUSTIFICATION

We thank the reviewer for bringing this to our attention. We measured GBV at both the work – and the interpersonal level and believe the reviewer may be referring to results from the interpersonal level, where indeed, GBV was more prevalent amongst PP recipients in 2013 and 2015. However, in the discussion, we refer to the prevalence of GBV at the work environment level, which was consistently higher amongst PP recipients amongst all years, hence, we do not deem it necessary to provide further evidence. Nonetheless, we have altered this paragraph to make it clearer that we are referring to GBV at the work level, rather than the interpersonal level. 

COMMENT REVIEWER 2

Besides, the authors showed that consistent condom use and GBV were both more prevalent among PP recipients, meanwhile suggested FSW experienced GBV may be less inclined to seek condoms. These two facts contradicted with each other. It could be great if the author could settle this contradiction that why PP recipient which experienced more GBV than non-recipients displayed more consistent condom use behavior.

CHANGES MADE

Please, see changes made to the Discussion.

JUSTIFICATION

We agree that our findings regarding the higher prevalence of GBV yet higher use of condoms amongst PP recipients is perplexing. We have added subsequent sentence, highlighting the need for future research to investigate this further, however, do not believe additional analysis falls within the scope of this research. 

COMMENT REVIEWER 2

Given the fact that the current design and analysis were inadequate for a causal reference, the statement in discussion: “sex without a condom may impede access to the PP” sounds a little inappropriate.

CHANGES MADE

Please, see changes made to the Discussion.

JUSTIFICATION

We thank the reviewer for their comment and have rephrased this sentence accordingly. 

COMMENT REVIEWER 2

Overall, this manuscript utilized longitudinal data from a series of national survey with about 5 waves. However, the “trend” of service coverage in this paper was not supported by testing for trends. It would be great if the author could provide more statistical evidence to support the description of coverage changes over the years.

CHANGES MADE

No changes made.

JUSTIFICATION

We appreciate the reviewer's suggestions here on how to improve the analysis.

In principle, we are open to conducting further statistical analyses, however our preference would be to not do so for two reasons: 1) the aim of the study was exploratory in nature, with the potential to suggest aspects for further study (with the appropriate methodology and statistical power) rather than to provide inferential estimates of relationships. Thus, we feel that the descriptive nature of the analyses allows for this exploration, without relying on inferential statistics to determine what is significant and what is not; 2) as described in the Discussion there were some concerns with the underlying data - we feel that the problems with the data do not preclude a rather descriptive and exploratory analysis, however conducting inferential statistics may suggest more confidence than we have in the assessed relationships.

*******

COMMENT REVIEWER 3

This paper has comprehensively described the access to HIV-prevention in female sex workers in Ukraine between 2009 and 2017: coverage, barriers and facilitators. 

CHANGES MADE

No changes made.

JUSTIFICATION

We thank the reviewer for their positive feedback.

COMMENT REVIEWER 3

While this article has provided numerous information including literature review, document analysis, exploratory analysis. The rich information is not easy to follow to generate a whole story to a specific research question. It has four objectives, which in my opinion is way too long for an article. The contents look more like a project report instead of one original research. I would suggest the authors to shorten the article and specify research questions instead of lump everything together and resubmit again.

CHANGES MADE

Please, see changes made in the Method section and the Supporting Information.

JUSTIFICATION

We agree with the reviewer that the manuscript is quite lengthy. However, we do deem each objective to be integral in painting and overall, clear picture of the current HIV PP coverage amongst Ukrainian FSWs between 2009 and 2017. As no there is no research with such breadth investigating this study aim, we believe our research to provide a valuable launching pad for future research. WE outline in the discussion that further research is needed, with more robust statistical testing to investigate select variables and our findings. Nonetheless, to reduce the Method section, we have removed the paragraph elaborating on the literature review details to the Supporting Information S1.

COMMENT REVIEWER 3

Line 82, the reference is not right.

CHANGES MADE

No changes made.

JUSTIFICATION

Perhaps because of a discrepancy in the line numbering, we were unable to locate an incorrect reference in line 82 (indeed there was no reference in line 82); however, we have now double checked that all references are correct

---

## [Decision Letter · Decision Letter 1]

30 Mar 2021

Access to HIV-prevention in female sex workers in Ukraine between 2009 and 2017: coverage, barriers and facilitators

PONE-D-20-24729R1

Dear Dr. Blumer,

We’re pleased to inform you that your manuscript has been judged scientifically suitable for publication and will be formally accepted for publication once it meets all outstanding technical requirements.

Kind regards,

Zixin Wang, PhD.

Academic Editor

PLOS ONE

Additional Editor Comments (optional):

Reviewers' comments:

Reviewer's Responses to Questions

**Comments to the Author**

1. If the authors have adequately addressed your comments raised in a previous round of review and you feel that this manuscript is now acceptable for publication, you may indicate that here to bypass the “Comments to the Author” section, enter your conflict of interest statement in the “Confidential to Editor” section, and submit your "Accept" recommendation.

Reviewer #1: (No Response)

Reviewer #2: All comments have been addressed

2. Is the manuscript technically sound, and do the data support the conclusions?

Reviewer #1: (No Response)

Reviewer #2: Yes

3. Has the statistical analysis been performed appropriately and rigorously? 

Reviewer #1: Yes

Reviewer #2: Yes

4. Have the authors made all data underlying the findings in their manuscript fully available?

Reviewer #1: No

Reviewer #2: Yes

5. Is the manuscript presented in an intelligible fashion and written in standard English?

Reviewer #1: Yes

Reviewer #2: Yes

6. Review Comments to the Author

Reviewer #1: My comments have been adressed very well by the authors and the manuscript has improved a lot. I have no further comments.

Reviewer #2: The authors addressed all my comments. Though it remained confusing that PP-recipients experienced higher gender-based violence as well as showed higher condom use behavior, the authors stated clearly that further study is warranted particularly for this phenomenon. Hopefully further research will be conducted to address this problem.

7. PLOS authors have the option to publish the peer review history of their article (what does this mean?). If published, this will include your full peer review and any attached files.

Reviewer #1: No

Reviewer #2: No

---

## [Editor Report · Acceptance letter]

6 Apr 2021

PONE-D-20-24729R1 

Access to HIV-prevention in female sex workers in Ukraine between 2009 and 2017: coverage, barriers and facilitators 

Dear Dr. Blumer:

I'm pleased to inform you that your manuscript has been deemed suitable for publication in PLOS ONE. Congratulations! Your manuscript is now with our production department. 

Kind regards, 

on behalf of

Professor Zixin Wang 

Academic Editor

PLOS ONE